# A CMOS Integrator-Based Clock-Free Time-to-Digital Converter for Home-Monitoring LiDAR Sensors

**DOI:** 10.3390/s22020554

**Published:** 2022-01-11

**Authors:** Ying He, Sung Min Park

**Affiliations:** 1Department of Electronic and Electrical Engineering, Ewha Womans University, Seoul 03760, Korea; heying0404@naver.com; 2Graduate Program in Smart Factory, Ewha Womans University, Seoul 03760, Korea

**Keywords:** CMOS, integrator, LiDAR, PDH, sensor, TDC

## Abstract

This paper presents a nine-bit integrator-based time-to-digital converter (I-TDC) realized in a 180 nm CMOS technology for the applications of indoor home-monitoring light detection and ranging (LiDAR) sensors. The proposed I-TDC exploits a clock-free configuration so as to discard clock-related dynamic power consumption and some notorious issues such as skew, glitch, and synchronization. It consists of a one-dimensional (1D) flash TDC to generate coarse-control codes and an integrator with a peak detection and hold (PDH) circuit to produce fine-control codes. A thermometer-to-binary converter is added to the 1D flash TDC, yielding four-bit coarse codes so that the measured detection range can be represented by nine-bit digital codes in total. Test chips of the proposed I-TDC demonstrate the measured results of the 53 dB dynamic range, i.e., the maximum detection range of 33.6 m and the minimum range of 7.5 cm. The chip core occupies the area of 0.14 × 1.4 mm^2^, with the power dissipation of 1.6 mW from a single 1.2-V supply.

## 1. Introduction

Recently, light detection and ranging (LiDAR) sensors have been exploited in various time-of-flight (ToF) applications, including unmanned autonomous vehicles that can drive safely by recognizing the surrounding environment with real-time three-dimensional (3D) mapping, e.g., Google self-driving cars with panoramic scan sensors [1]; home-monitoring LiDAR sensors for either single elders living alone or senile dementia patients residing in nursing homes [2,3]; and remote sensing LiDAR sensors for the purpose of observation of forestry, the cryosphere, aerosols, and clouds.

In particular, home-monitoring LiDAR sensors have become urgently needed to help professional nurses take care of their dementia patients more efficiently. It is well known that many developed countries have been facing the surge of senior citizen populations due to their advanced health systems [2], which naturally leads to the proliferation of senile dementia patients. In South Korea, the Ministry of Health-Welfare predicted that 10% of senior citizens, i.e., one million elders, might suffer dementia in 2025. These elders would frequently suffer from unpredictable falling accidents, thus demanding extremely high social costs for their medical care. However, it is almost impossible to detect these falling accidents in real-time, especially when we consider the situations of senile dementia patients living alone at home and those staying in nursing homes where only one or two professional nurses reside and take care of 10–20 patients simultaneously. Even for those living with their families, it could be barely feasible. Therefore, it would be an efficient solution to equip home-monitoring LiDAR sensors in houses and at nursing homes to ensure that these accidents can be detected instantly and alarms can be conveyed to either nurses or families in real-time, hence enabling prevention of more seriously dangerous incidents.

For this purpose, ToF-based pulsed LiDAR sensors are preferred because they can emit and receive light signals to measure the distance between its optical source and targets within the detection range. Additionally, they can compute the speed of moving targets by estimating the distance variation from two transmitted pulses [4]. Yet, these indoor environment perception LiDAR sensors mandate a fast response (>1000 results/s) so that the latency of the receiver circuit might become a key factor to limit the operation speed. Additionally, the dynamics of the received echo can be wide (>1:1000) even with a single transmitted pulse for targets located within the range of tens of meters [5].

Figure 1 illustrates the block diagram of a pulsed ToF LiDAR sensor where a laser transmits short light pulses to distant targets and a START pulse is concurrently sent to a time-to-digital converter (TDC) in the receiver (Rx), thereby initializing the measurement of the range detection [6]. The reflected optical pulses from targets are detected by the sensitive Rx, where the optical detector (typically avalanche photodiodes) converts the incoming optical signals into electrical currents. Then, these current pulses are amplified to output voltages by the front-end analog circuitry that consists of a transimpedance amplifier (TIA) and a post-amplifier (PA) [7]. Thereafter, the amplified output voltages (as known as STOP pulses) enter the TDC, where the time difference between the START pulse and the STOP pulse is estimated. Finally, the corresponding digital codes for the time interval are generated at the output of the TDC. Here, it is noted that the resolution of the TDC is determined by the minimum detection range, whereas the accuracy is affected by the quantization error caused by the rise time of the laser pulses and the delay of the receiver circuits [8]. In addition, the maximum detection range is directly relevant to how large the input dynamic range of the TDC circuit is. Nevertheless, due to the jitter accumulation and mismatch-related non-linearities, it is difficult to achieve a large input dynamic range while obtaining fine resolution characteristics [9].

Various architectures of TDCs have been introduced, including the one-dimensional (1D) flash TDC, two-dimensional (2D) Vernier TDC, three-dimensional (3D) Vernier TDC, cyclic Vernier TDC, gated ring oscillator (GRO) TDC, and time-amplifier (TA) TDC. These conventional TDCs can hardly achieve good timing accuracy due to a number of inherent issues such as walk-error, multiple echoes, metastability, etc. [10]. Hence, in order to achieve a large input dynamic range and good timing accuracy performance, we propose a novel nine-bit clock-free integrator-based TDC (I-TDC) which can detect targets within the range between several centimeters to a few tens of meters, with a very fast conversion rate.

Section 2 describes conventional TDCs and introduces the architecture of the proposed I-TDC. Section 3 shows its operations and the simulation results. Section 4 demonstrates the measured results. Then, the conclusion follows in Section 5.

## 2. Integrator-Based TDC

### 2.1. Conventional TDCs

A 1D flash TDC is the simplest configuration (as shown in Figure 2), in which a START pulse passes through a delay chain and then each delayed pulse samples a STOP pulse in each D-flipflop. Thereby, 15-bit thermometer output codes can be generated through this sampling and be finally converted to binary codes. Although this 1D flash TDC shows high conversion speed, its resolution is significantly vulnerable to process–voltage–temperature (PVT) variations [11]. Besides, it requires a large chip area for the purpose of long-range detection because more delay cells and DFFs should be added, hence limiting its usage to centimeter-level short range applications.

Vernier TDCs (2D and 3D) can achieve a sub-gate delay resolution with improved linearity. Since the START and the STOP signals propagate with different delays (e.g., τ_1_ and τ_2_), the resolution can be determined by the difference of these two delays (i.e., τ_1_–τ_2_), thus enabling us to achieve higher resolution. In addition, the 1st-order mismatches occurring in delay cells can be cancelled [12]. However, they require a large chip area because of the doubled number of delay cells. Additionally, they can scarcely avoid large dead-time that is limited by the total delay time [13].

Cyclic Vernier TDC and GRO TDC utilize counters and can achieve both fine resolution and large input range. The former exploits two voltage-controlled oscillators (VCOs) and counters. When a START pulse arrives, a slow VCO oscillates and a coarse counter counts the rising edges of the START pulse. When a STOP pulse enters, the coarse counter stops. Then, a fast VCO oscillates and a fine counter counts the number of cycles it takes for the fast VCO to catch up the slow VCO. However, this cyclic Vernier TDC suffers from high switching noise, large dynamic power consumption, the risk of metastability, and a large latency of the fine step measurement [14]. The latter employs a ring oscillator in which the phase difference between each node of the oscillator is utilized to achieve high resolution and wide detection range. Then, the time interval between START and STOP pulses is acquired by measuring the number of delay element transitions with a counter. Although it can yield good noise characteristics by the 1st-order noise shaping, the circuit structure is quite complicated [15]. These counter-based TDCs mandate fast clock signals to achieve high resolution, which, however, may lead to severe quantization error.

TA TDC provides fine-time resolution and a high conversion rate with residue time [16]. Yet, it requires large gain and can be unstable at wide time differences, thus requiring an extra calibration scheme. Besides, it suffers from its limited detection range and high power consumption. Table 1 summarizes the comparison of the I-TDC with others.

### 2.2. Proposed I-TDC

In order to overcome the aforementioned issues of conventional TDCs, we propose a novel clock-free integrator-based TDC (I-TDC). Figure 2 depicts the block diagram of the proposed nine-bit I-TDC that combines a 1D flash TDC for coarse control [17] and an integrator charging circuit with a peak–detect–hold circuit for fine control [18], hence enabling us to provide fine resolution, wide detection range, low dynamic power consumption, and fast conversion rate characteristics. A thermometer-to-binary converter (not shown in Figure 2 for simplicity) can be added to the 1D flash TDC for the generation of four-bit coarse-control codes such that the measured detection range can be represented by nine-bit digital codes in total.

The basic concept of the proposed I-TDC is to convert a time interval into a voltage, which is similar to a time-to-voltage converter (TVC) [19]. Therefore, the I-TDC exploits an integrator in which a capacitor is charged for the duration of a time interval and generates a corresponding voltage. In order to improve the linearity of this circuit, a resistor can be added before the charging capacitor. Additionally, a NMOS switch can be added to discharge extra charges from the capacitor, hence effectively reducing the undesired offset voltages. The following peak detection and hold (PDH) circuit holds the peak voltage for a far longer time during which the fine-control measurements can be facilely conducted. Here, it should be noted that this I-TDC employs a clock-free configuration. Hence, not only dynamic power consumption can be reduced but also there is no need to synchronize START and STOP pulses with clock signals that otherwise often give rise to different rising/falling edges.

As described above, the total time interval between START and STOP pulses are represented by a nine-bit output digital code, i.e., a four-bit coarse-control code and a five-bit fine-control code. Post-layout simulations confirm that the maximum detection range can reach 36 m, which corresponds to a 120 ns time interval with the resolution (τ) of 8 ns, i.e., τ_1_ − τ_2_ = 8 ns, and also that the minimum detection distance can be as short as 7.5 cm, which corresponds to a 0.25 ns time interval.

#### 2.2.1. Flash TDC

The exploited 1D flash TDC enables high-speed timing, single-event timing measurements, and simple digital implementation. Figure 2 shows the block diagram of the 1D flash TDC which incorporates a delay chain and a number of D-FF arrays to generate 15 pulses. X is the output signal of the edge detector (ED) and is generated by detecting the falling edges of both START and STOP pulses. Thereafter, it enters series CMOS buffer delay lines. As the X signal passes through the serial elements, it captures (or samples) STOP pulses at every 8 ns delay.

In this coarse-control scheme, the START pulses are delayed by an integer multiple of a buffer delay. Then, each delay signal samples the STOP pulses in D-FFs, hence generating 15 output signals which can be translated to a 120 ns maximum time interval. Therefore, the detection range can reach 36 m. Then, a thermometer-to-binary converter is needed to provide final four-bit binary digital codes from these coarse 15 pulses.

#### 2.2.2. Integrator Charging Circuit

Figure 3 shows the schematic diagram of the proposed integrator charging circuit that generates the output current (I_X_ = 60 μA) when an input pulse (Q_X_) enters. Q_X_ is an inverted signal of each coarse output C_X_ (X = 1~15) by using an inverter with small delay. Then, it yields an output voltage (V_X_) at a load capacitor (C_Load_ = 0.4 pF), which is given by,
(1)$$V_{X}=I_{X}\frac{\tau }{C_{\mathrm{Load}}}$$
where τ represents the pulse width of Q_X_.

In this work, an OP-AMP is preferred to a simple passive switch for more effective conversion from an input pulse to an output voltage. Additionally, the OP-AMP incorporates PMOS input transistors for low flicker noises. A series resistor (R_0_ = 3.4 kΩ) is inserted to isolate C_Load_ from the output of the OP-AMP, which helps to enhance the linearity of this integrator charging circuit. In addition, an NMOS switch (Q_0_) is connected in parallel with C_Load_ so as to remove not only the DC offset voltages but also the remaining charges in C_X_.

#### 2.2.3. PDH Circuit

Figure 3 also depicts the schematic diagram of the peak-detect-hold (PDH) circuit, in which a short input pulse (V_X_) arrives from the preceding integrator charging circuit. Then, a sharp negative transition occurs at the output of the OP-AMP, which will turn on the PMOS pass-transistor (Q_0_) and start to charge the output capacitor (C_OUT_ = 1 pF).

When V_X_ reaches the peak value, F_X_ will become the maximum voltage. With no DC path to discharge C_OUT_, this maximum voltage will maintain until the reset (RST) transistor is turned on. With a high RST signal, C_OUT_ will be discharged and accordingly F_X_ will go down to GND, resetting the operations.

## 3. Operation and Simulation Results

### 3.1. Operation Modes

The proposed I-TDC operates in three different modes. Figure 4 shows the conceptual waveforms of these three different modes. First, Mode 1 is the reset mode, which begins when a START pulse enters. During this mode, all the outputs are reset to GND and then are prepared for the next arrival of input signals. Therefore, the START and STOP pulses can be read in this mode. Second, Mode 2 is the charging mode that begins when the STOP signal goes high. During this mode, the 1D flash TDC generates coarse codes. Both C_Load_ and C_OUT_ are charged to certain voltage levels that are proportional to the pulse widths of the corresponding coarse codes. Third, Mode 3 is the holding mode, which begins when the STOP signal goes low. During this mode, all the coarse codes are refreshed to GND. Yet, the voltages charged at C_Load_ and C_OUT_ maintain until a next reset mode begins.

Figure 4 also illustrates an example of these various waveforms for a time interval of 21 ns between START and STOP pulses. In this case, the coarse-control codes (C1 and C2) generate 8 ns pulses, while C3 yields a 5 ns pulse. These pulses are inputted to charge C_Load_ during the pulse width of each pulse at the same slope of 1.2 V/8 ns. Then, V1 and V2 are charged up to 1.2 V (=VDD), whereas V3 is charged to 0.75 V. After fine tuning, the final output peak voltages (F1 and F2 = 1.2 V, and F3 = 0.75 V) can be maintained by the PDH circuits until a next input signal arrives.

### 3.2. Simulation Results

Figure 5 shows the simulation results of the proposed I-TDC for a number of time intervals between START and STOP pulses, in which the output voltages (F1–F15) are generated. First, for the case of no time interval, as shown in Figure 5a, the minimum voltage of 10 mV is detected at all the output nodes.

Figure 5b shows the simulated output voltages for a 0.25 ns time interval between START and STOP pulses, which corresponds to the target distance of 7.5 cm. Then, F2–F15 detect the minimum voltage of 10 mV, whereas F1 only is charged to 33 mV. These result in the estimated time interval of 0.22 ns.

Figure 5c depicts the resulting output voltages for a 49 ns time interval that corresponds to a 14.7 m distance to target. In this situation, F1–F6 are charged up to the maximum voltage of 1.197 V, whereas F7 is charged to 0.158 V and F8–F15 lie at the minimum voltage of 0.015 V. These results lead to the almost identical time interval of 49.05 ns.

Figure 5d illustrates the resulting output voltages for a 95 ns time interval between START and STOP pulses, which corresponds to the target distance of 28.5 m. Therefore, F1–F11 are charged up to the maximum voltage of 1.197 V, whereas F12 is charged to 1.049 V and F13–F15 lie at the minimum voltage of 0.015 V. These result in the identical time interval of 95 ns.

Figure 5e depicts the resulting output voltages for a 119.75 ns time interval, which corresponds to a 35.93 m distance to target. Here, F1–F14 are charged up to the maximum voltage of 1.197 V, whereas F15 is charged to 1.17 V. These indicate an almost identical 119.8 ns time interval.

Figure 5f shows the resulting output voltages for a 120 ns time interval that corresponds to the target distance of 36 m. All the outputs are charged to the maximum voltage of 1.196 V. These confirm the estimation of a 120 ns time interval.

## 4. Discussion

### 4.1. Chip Implementation

Figure 6 shows the chip microphotograph of the proposed I-TDC realized in a 180 nm CMOS technology and its test setup, where the chip core occupies the area of 0.14 × 1.4 mm^2^. DC measurements reveal that the I-TDC consumes 1.6 mW from a single 1.2-V supply. For pulse measurements, a waveform generator (AFG31252, Tektronix, Beaverton, OR, USA) was utilized to provide two pulses with small phase difference between the START and the STOP pulses. The parallel output voltages of the I-TDC were sampled by using the analog mode of a logic analyzer (Logic Pro16, Saleae, San Francisco, CA, USA).

### 4.2. Measured Results

Figure 7 demonstrates the measured output pulses of the I-TDC for various time intervals of 0.5 ns, 8 ns, 49 ns, 95 ns, 111.5 ns, and 112 ns. These correspond to the detection range of 15 cm~33.6 m. For these measurements, two threshold voltages are selected, i.e., one (V_H_) is set to ~950 mV and the other (V_L_) is set to the LSB, which is 37.5 mV. Then, the results (F1–F15) of the I-TDC are compared with V_H_, resulting in a coarse thermometer code that will be eventually converted to a binary code. Thereafter, a result in between V_H_ and V_L_ are selected among F1–F15 and connected to a simple five-bit ADC by using a switch, finally generating a fine-control code. It is noted here that only a five-bit ADC with 15 switches connecting F1–F15 can be utilized for this proposed I-TDC, hence enabling us to avoid the undesired increase of power consumption.

Figure 7a shows the resulting output voltages for a 0.5 ns time interval between START and STOP pulses, which corresponds to a 15 cm distance to target. The measured output pulse of ~0 V is detected from F2 to F14. Only F1 is charged with the voltage difference (ΔV) of 73 mV. Therefore, a coarse-control code with 14 zeros is generated and F1 is connected to the five-bit ADC, producing a five-bit fine code (i.e., 00010). These represent the time interval of 0.57 ns.

Figure 7b is the measured output voltages for a 8 ns time interval between START and STOP pulses, which corresponds to the target distance of 2.4 m. The minimum voltage of ~0 V is detected from F2 to F14. Only F1 is charged to ~1.04 V. Thus, a coarse-control code with 1 one and 13 zeros is generated. These result in the estimated time interval of 8.32 ns.

Figure 7c is the measured output voltages for a 49 ns time interval that corresponds to a 14.7 m distance to target. Here, F1–F6 are charged to ~1.0 V, whereas the minimum output voltage of ~0 V is detected from F8 to F14, and F7 is charged with the voltage difference (ΔV) of 94 mV. Therefore, a coarse-control code with 6 ones and 8 zeros is generated, and F7 is connected to the five-bit ADC, producing a five-bit fine code (i.e., 00010). These lead to the almost identical time interval of 48.93 ns.

Figure 7d is the resulting output voltages for a 95 ns time interval, which corresponds to the target distance of 28.5 m. It is clearly seen that F1–F11 are charged to ~1.0 V, whereas F12 yields the voltage difference of 845 mV and F13–F14 are at the minimum voltage of 0 V. Thus, a coarse-control code with 11 ones and 3 zeros is generated, and F12 is connected to the five-bit ADC, producing a five-bit fine code (i.e., 10110). These indicate the estimated time interval of 94.87 ns.

Figure 7e demonstrates the measured output voltages for a 111.5 ns time interval, which corresponds to a 33.45 m distance to target. Here, F1–F13 are charged up to ~1.0 V, whereas F14 shows the voltage difference of 811 mV. Therefore, a coarse-control code with 13 ones and 1 zero is generated, and F11 is connected to the five-bit ADC, producing a five-bit fine code (i.e., 10101). These results represent the estimated time interval of 111.31 ns.

Figure 7f is the finally measured output voltages for a 112 ns time interval between START and STOP pulses, which corresponds to a 33.6 m distance to target. It is clearly seen that F1–F13 are charged up to ~1.0 V, whereas F14 is charged to 0.99 V with the voltage difference (ΔV) of 888 mV. Thus, a coarse-control code with 14 ones is generated, indicating the maximum range detection. These lead to the estimated time interval of 111.92 ns.

Here, it is noted that only 14-bit data outputs were acquired in these measurements because of the limitation of the logic analyzer (Saleae Logic Pro16). Yet, five-bit fine-control digital codes can be obtained with a five-bit ADC that accounts for 0.5 ns TDC resolution. Provided that 15-bit data outputs could be achieved, the farthest detection range would reach 36 m, thus resulting in the feasible dynamic range of 53.6 dB.

Table 2 compares the performance of the proposed I-TDC with the previously reported TDCs specialized for LiDAR applications. In [20], a cyclic Vernier TDC was presented, demonstrating a fine detection depth of 0.57 cm and the maximum range of 53.2 m. However, the measured dynamic power was rather large for the 0.67-MS/s conversion rate. In [21], a 3D modified Vernier TDC was realized to achieve 8-bit digital output codes. Nonetheless, the conversion rate was limited to 50 MS/s and the minimum detection range was rather large. In [22], an 11-bit 3D Vernier TDC was demonstrated with very low power dissipation. Yet, the formidable clock issues could not be avoided and the maximum detection range was limited to 4.3 m. In [23], an 11-bit GRO TDC achieved the maximum detection range of 48 m, in which, however, the power consumption was as extremely large as 180 mW. In [24], a 15-bit resolution TDC was realized with the widest dynamic range at a rather moderate power of 5.2 mW. Yet, it mandated a 9.4-bit ADC, hence leading to undesirable circuit complexity.

## 5. Conclusions

We have realized a nine-bit I-TDC for the applications of indoor home-monitoring LiDAR sensors, in which a 1D flash TDC generates coarse digital pulses that are thereafter injected into an integrator with a PDH circuit, which facilitates the generation of fine resolution. Since the proposed I-TDC exploits a clock-free configuration, it can reduce dynamic power consumption and discard serious clock-related issues such as glitch, skew, and switching noise. Test chips implemented by using a 180 nm CMOS process demonstrate the measured dynamic range of 53 dB, i.e., the detection range of 7.5 cm to 33.6 m which corresponds to the time intervals of 0.5–112 ns. Hence, it can be concluded that the proposed I-TDC provides a feasible low-cost, low-power solution for indoor home-monitoring LiDAR sensors.

## Figures and Tables

**Figure 1 sensors-22-00554-f001:**
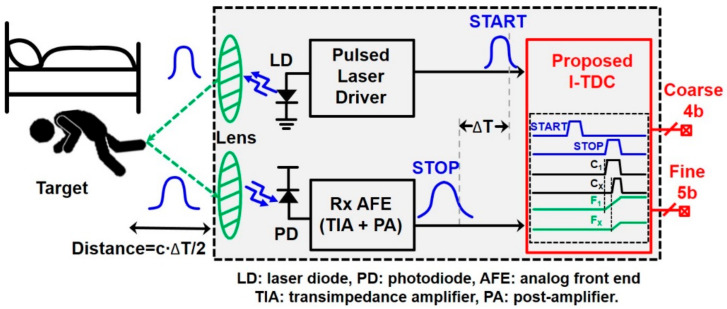
Block diagram of a typical LiDAR sensor.

**Figure 2 sensors-22-00554-f002:**
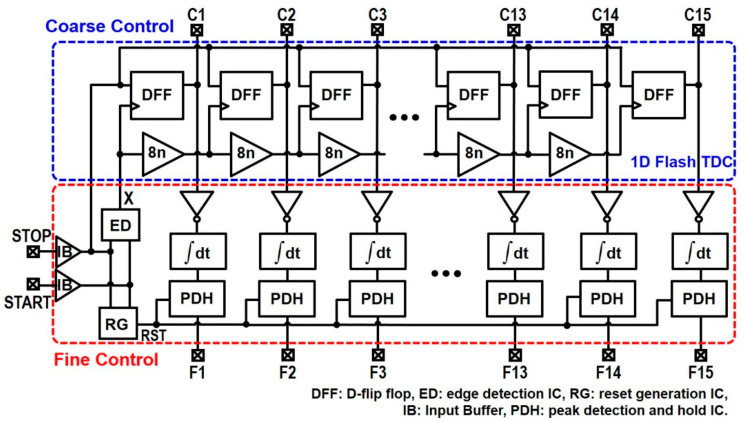
Block diagram of the proposed nine-bit I-TDC.

**Figure 3 sensors-22-00554-f003:**
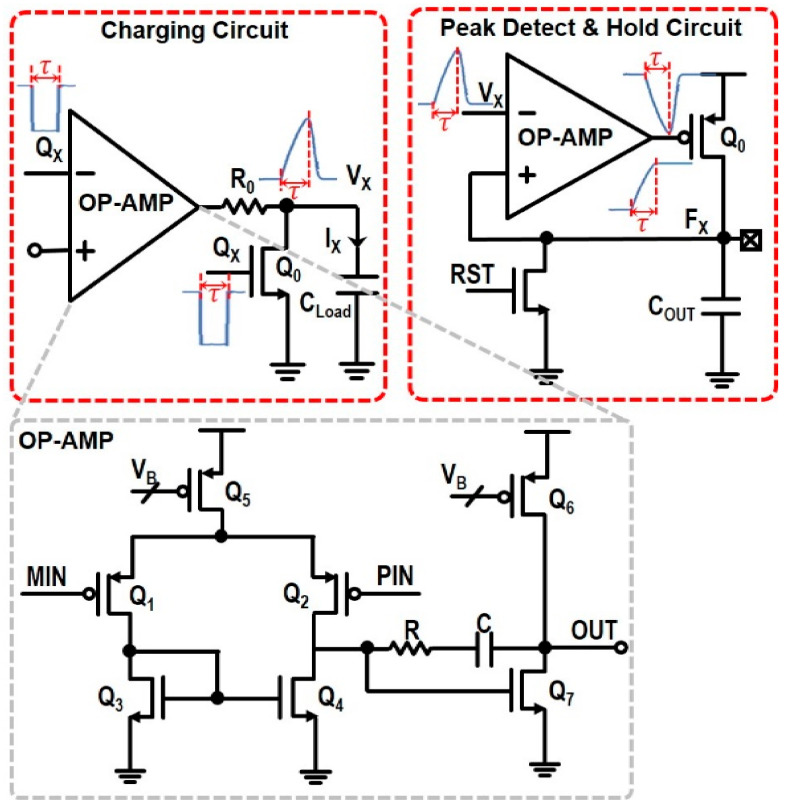
Schematic diagrams of the integrator charging circuit and the PDH circuit.

**Figure 4 sensors-22-00554-f004:**
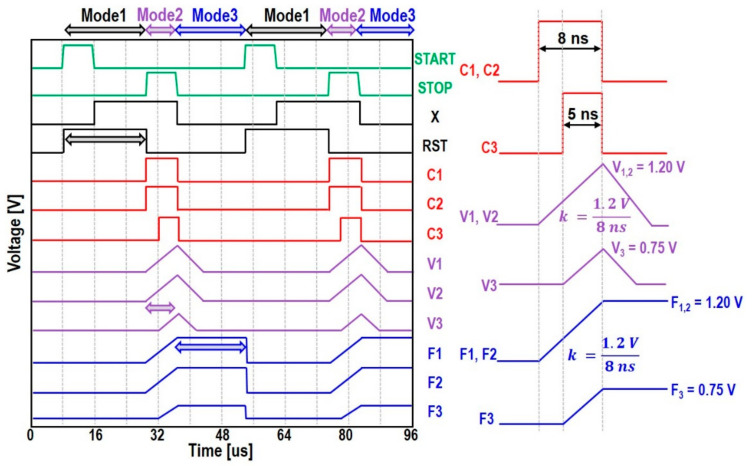
An example of various waveforms for three different modes.

**Figure 5 sensors-22-00554-f005:**
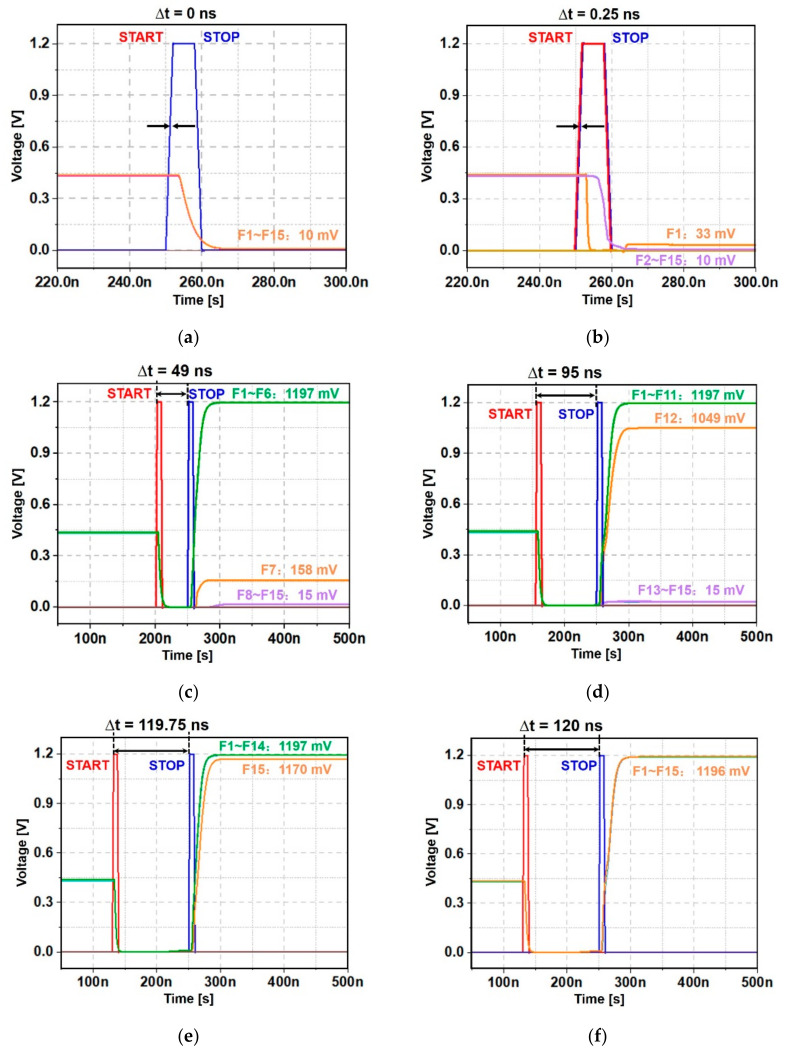
Simulated results of the proposed I-TDC for (**a**) no time interval, (**b**) 0.25 ns, (**c**) 49 ns, (**d**) 95 ns, (**e**) 119.75 ns, and (**f**) 120 ns time intervals.

**Figure 6 sensors-22-00554-f006:**
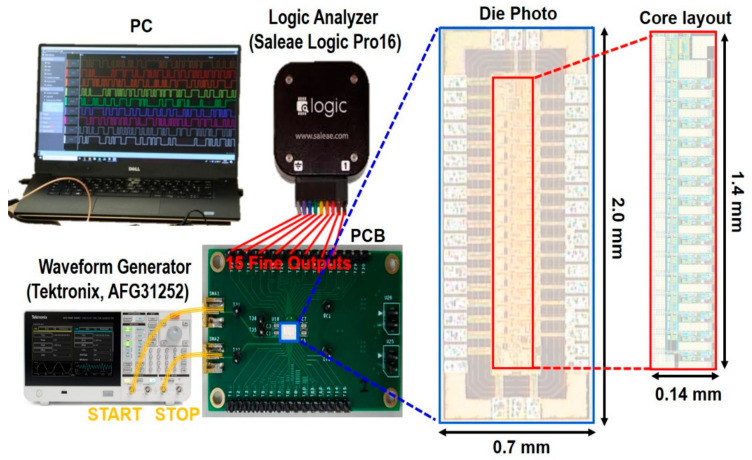
Chip photograph of the I-TDC and its test setup.

**Figure 7 sensors-22-00554-f007:**
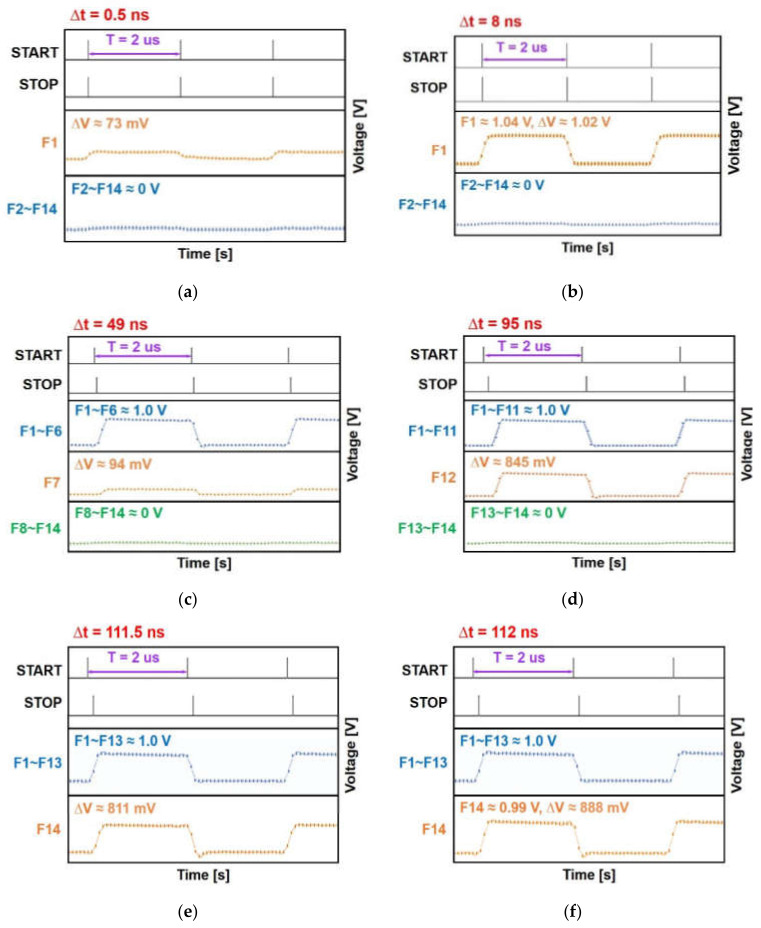
Measured results of the proposed I-TDC for (**a**) 0.5 ns, (**b**) 8 ns, (**c**) 49 ns, (**d**) 95 ns, (**e**) 111.5 ns, and (**f**) 112 ns time interval, respectively.

**Table 1 sensors-22-00554-t001:** Comparison of the proposed I-TDC with other TDCs.

TDCs	Resolution	Conversion Rate	Max. Detection Range	Disadvantages
Flash	coarse	high	small	vulnerable to PVT variations and large chip area
Vernier	fine	low	moderate	large chip area and long dead time
Cyclic Vernier	fine	very low	large	severe switching noise, high dynamic power consumption, and large latency
GRO	fine	low	large	fast clock required and severe quantization error
TA	fine	high	large	high power consumption and large gain required with extra calibration
Proposed I-TDC	fine	high	large	sensitive to PVT variations

GRO: gated ring oscillator and TA: time-amplifier.

**Table 2 sensors-22-00554-t002:** Performance comparison of the proposed I-TDC with prior arts.

Parameters	[20]	[21]	[22]	[23]	[24]	This Work
CMOS (nm)	180	130	130	110	65	180
Configuration	Cyclic Vernier	3D	3D	GRO	ADC × TDC	Two-step
Supply voltage (V)	1.8	1.2	1.5	1.5	1.2	1.2
Resolution (bit)	9.88	8	11	11	15	9
LSB (ps)	377	625	6.98	156.25	6.25	500
Conversion rate (MS/s)	0.67	50	25	100	500	4,000
Detection range	0.57 cm~53.2 m	18.8 cm~28.8 m	2.1 mm~4.3 m	2.34 cm~48 m	1.83 mm~60 m *	7.5 cm~33.6 m
Power dissipation (mW)	0.65	2.9	0.33	180	5.2	1.6
FoM (pJ/conv.-step)	1.030	0.227	0.064	0.879	0.0003	0.0008
Core area (mm^2^)	0.028	0.33	0.28	29.75	0.06	0.20

* estimated from the time resolution multiplied by 2^N^ and the speed of light FoM = Power/(2^N^ ∗ F_S_) [pJ/conversion-step].

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
