# Peer review of "A CMOS Integrator-Based Clock-Free Time-to-Digital Converter for Home-Monitoring LiDAR Sensors"

_sensors, 2022, doi:10.3390/s22020554_

Round 1

Reviewer 1 Report

The authors propose an integrator-based clock-free TDC. The coarse count is generated by delay chain structure, and the fine count is generated by an integrator charging circuit followed by a peak-detect-hold circuit. It is innovative but lacks practical value.

  1. It is said that 1-D flash TDC’S resolution is significantly vulnerable by PVT in the a. The structure of this article is based on this. So how much is the influence of PVT on TDC in the article?
  2. The article mentions a thermometer-to-binary converter is needed, but the structure is not shown, please complete it.
  3. There is a picture mark error in the text, Fig.5 in line 175 should be Fig.4, please correct it.

Author Response

Reviewer: 1

The authors propose an integrator-based clock-free TDC. The coarse count is generated by delay chain structure, and the fine count is generated by an integrator charging circuit followed by a peak-detect-hold circuit. It is innovative but lacks practical value.

1. It is said that 1-D flash TDC’s resolution is significantly vulnerable by PVT in the a. The structure of this article is based on this. So how much is the influence of PVT on TDC in the article?

-> As commented, the proposed TDC would be influenced by PVT variations. Particularly, the unit delay-cell in the 1-D flash TDC can be significantly vulnerable to the PVT variations. Therefore, we have checked it for three worst conditions and then, the post-layout simulations confirm that each unit delay shows the largest deviation of 0.7 ns from its typical delay of 8 ns. So, the charged peak voltage would be different, which is a defect of this I-TDC as described in Table 1 (page 3) in the revised manuscript.

2. The article mentions a thermometer-to-binary converter is needed, but the structure is not shown, please complete it.

-> We apologize for this confusion. Certainly, a conventional 4-bit thermometer-to-binary converter can be added to obtain coarse control codes. The thermometer-to-binary converter is not shown in Fig. 2 for simplicity of the whole block diagram. We have revised the sentence (page 3, line 130-133) as below, “A thermometer-to-binary converter (not shown in Fig. 2 for simplicity) can be added to the 1D flash TDC for the generation of 4-bit coarse control codes, such that the measured detection range can be represented by 9-bit digital codes in total.”

3. There is a picture mark error in the text, Fig.5 in line 175 should be Fig.4, please correct it.

-> We have corrected it.

Reviewer 2 Report

The paper’s scope entitled “A CMOS Integrator-based Clock-free Time-to-Digital Converter for Home-Monitoring LiDAR Sensors” is concerned with developing a time-to-digital converter (TDC) using integrators without a clock generator for indoor monitoring applications. The results indicate that the integrators may convert the time interval into the voltage, and this paper’s technical approach is attractive from developing clock-free low power TDC. However, some technical details are unclear and need to be addressed to convince this research. I concluded that the paper should be rejected in the MDPI Sensors.

  1. As shown in Fig. 8, the proposed circuit requires measuring the analog voltage of the fine output (F1) to detect 0.5-nanosecond interval. That is quite strange because the output must be digital values. ADC or a comparator is required for each fine output terminal. Therefore, it seems that the fabricated circuit is not a completed TDC. Also, the core area and power dissipation shown in Table 1 do not consider these digital-converting parts. It is not a fair comparison.
  2. The role of the fine output and coarse output is not unclear. Why are the coarse outputs required? In the case of the Vernier TDC, fine output compensates the flash TDC output by remeasuring using the second TDC with better resolution. On the other hand, the fine output in the proposed circuit might not have a better resolution than the first flash TDC if the output is fully digitalized.
  3. The author should describe the theoretical minimum resolution and why the fine measurement system can measure the input interval better than the coarse measurement (1D flash TDC). Besides, the resolution of the fine control may depend on Cload and Ix­­. What are the required conditions of these parameters?
  4. Due to the principle, 2^n-1buffers, DFF circuits, integrators, and peak-detect&hold circuits are required in the proposed circuits for n-bit TDC. However, the author describes that the proposed circuit can save a larger area than the conventional flash TDC or Vernier TDC. Why?
  5. Introduction: Why dToF measurement is required? iTOF (AMCW) and FMCW LiDAR methods can also be used. What is the advantage of using ToF in home-monitoring LiDAR?

Author Response

Reviewer: 2

The paper’s scope entitled “A CMOS Integrator-based Clock-free Time-to-Digital Converter for Home-Monitoring LiDAR Sensors” is concerned with developing a time-to-digital converter (TDC) using integrators without a clock generator for indoor monitoring applications. The results indicate that the integrators may convert the time interval into the voltage, and this paper’s technical approach is attractive from developing clock-free low power TDC. However, some technical details are unclear and need to be addressed to convince this research. I concluded that the paper should be rejected in the MDPI Sensors.

1. As shown in Fig. 8, the proposed circuit requires measuring the analog voltage of the fine output (F1) to detect 0.5-nanosecond interval. That is quite strange because the output must be digital values. ADC or a comparator is required for each fine output terminal. Therefore, it seems that the fabricated circuit is not a completed TDC. Also, the core area and power dissipation shown in Table 1 do not consider these digital-converting parts. It is not a fair comparison.

-> Thanks a lot for your concern. As commented, either ADCs or comparators are needed to generate digital codes. However, in our test setup, the analog outputs are connected to a logic analyzer s shown in Fig. 7, which then results in the corresponding digital outputs. Nonetheless, according to our previous experience, we reckon that the power consumption would not be quite different from the value shown in Table 2, because comparators do not contribute power consumption as anticipated. But the core area would be increased, i.e., roughly 0.1 mm2 larger with a 5-bit SAR ADC for fine-control codes.

2. The role of the fine output and coarse output is not unclear. Why are the coarse outputs required? In the case of the Vernier TDC, fine output compensates the flash TDC output by remeasuring using the second TDC with better resolution. On the other hand, the fine output in the proposed circuit might not have a better resolution than the first flash TDC if the output is fully digitalized.

-> In the proposed I-TDC, the coarse outputs have the resolution of 8 ns that corresponds to the detection range of 1.2 meters. Thereafter, these coarse outputs are inserted into the fine control circuit so as to finally generate the outputs of which resolution can be 0.5 ns that corresponds to 7.5 centimeters.

3. The author should describe the theoretical minimum resolution and why the fine measurement system can measure the input interval better than the coarse measurement (1D flash TDC). Besides, the resolution of the fine control may depend on Cloadand Ix­­. What are the required conditions of these parameters?

-> The theoretical minimum resolution of this I-TDC is 0.25 ns.

-> The 1D flash TDC has the limit of 8-ns time interval in this work. Therefore, it cannot measure the time interval of less than 8 ns. However, the fine measurement utilizes the linear analog output voltages and hence can measure more precisely and almost linearly the time interval longer than 0.25 ns.

-> As commented, the fine resolution of this I-TDC is dependent upon the linearity of the charging circuit, the values of Cload (0.4 pF) and Ix (= 60 uA) were carefully selected under the considerations of pulse width and desired peak voltage.

4. Due to the principle, 2^n-1buffers, DFF circuits, integrators, and peak-detect&hold circuits are required in the proposed circuits for n-bit TDC. However, the author describes that the proposed circuit can save a larger area than the conventional flash TDC or Vernier TDC. Why?

-> A conventional flash TDC cannot easily acquire the same sub-gate-delay time resolution as the proposed I-TDC. For this purpose, too many DFFs are required in a flash TDC.

-> Vernier TDCs can achieve finer resolution than a flash TDC. Still, they need 480 arbiters (or 36 x 15 = 540 DFFs for example) to detect 120-ns time difference with 0.25-ns resolution, surely requiring larger chip area for the same detection range than this I-TDC.

5. Introduction: Why dToF measurement is required? iTOF (AMCW) and FMCW LiDAR methods can also be used. What is the advantage of using ToF in home-monitoring LiDAR?

-> We don’t keep dToF to ourselves for the applications of home-monitoring sensors. As commented, iToF and FMCW LiDARs can be applied as well. The former is more useful in short-range and thus more suitable for home-monitoring LiDAR sensors. The latter is more popular for the applications of self-driving cars, i.e., for longer range detection. Yet, dToF has been very well-known and simple to build up, which compelled us to utilize it for now. However, readers may choose iToF method for these home-monitoring sensors.

Reviewer 3 Report

Page 1, line 13. Omit “also”.

Page 1, line 30. Omit “own”.

Page 1, line 41. Omit “for those over 65”. It is indicated already at the beginning of this sentence.

Introduction, second paragraph. It can be shortened referring to one country only (“For example…”).

Page 2, lines 63-65. You do not show these blocks, so the whole sentence can be omitted. 

Page, line 65. Not “enlarged”, but “amplified”.

 Fig. 2 and Fig. 3 are given in the wrong order, and do not correspond to the description order. Please clarify the whole part 2.1.

Page 3, line 93. Change “it cannot avoid large chip area occupation” into “requires large chip area”. Will be direct and clear. Besides, it is not clear why large area limits the range. Explain.

Page 3. Description of other TDCs, line 96-120. It is better to give the comparative table giving comparison of your converter with others. If you do not want to do this, just eliminate this material. And I did not find real discussion of the comparison table later on.

Page 5, line 175. You write “Fig. 5 depicts the schematic diagram…”. Fig. 5 depicts the waveforms. So, please, put the figures in correspondence with the text. You had the same problem with Fig. 2 and Fig. 3.

Page 9, Fig. 8. This figure describes the results of measurements, not the results of simulation, as I am supposing.

Simplify the style and improve the grammar.

Author Response

Reviewer: 3

Page 1, line 13. Omit “also”.

-> It is omitted.

Page 1, line 30. Omit “own”.

-> It is omitted.

Page 1, line 41. Omit “for those over 65”. It is indicated already at the beginning of this sentence.

-> It is omitted.

Introduction, second paragraph. It can be shortened referring to one country only (“For example…”).

-> Thanks a lot for your concern. As you may know, we authors were trying to explain the global situations especially in Far East, Europe, and North America. Therefore, we believe that this global need would grasp many more readers’ attention. So, we sincerely hope that this paragraph would not bother the reviewer too much.

Page 2, lines 63-65. You do not show these blocks, so the whole sentence can be omitted.

-> We have modified Fig. 1 so as to show (TIA + PA) included in Rx.

Page, line 65. Not “enlarged”, but “amplified”.

-> It is changed.

 Fig. 2 and Fig. 3 are given in the wrong order, and do not correspond to the description order. Please clarify the whole part 2.1.

-> We have merged Fig. 3 to Fig. 2 for clarity.

Page 3, line 93. Change “it cannot avoid large chip area occupation” into “requires large chip area”. Will be direct and clear. Besides, it is not clear why large area limits the range. Explain.

-> It is changed to “Besides, it requires large chip area for the purpose of long-range detection because more delay cells and DFFs should be added, hence limiting its usage to centimeter-level short range applications.”

Page 3. Description of other TDCs, line 96-120. It is better to give the comparative table giving comparison of your converter with others. If you do not want to do this, just eliminate this material. And I did not find real discussion of the comparison table later on.

-> Thanks for your comment. We have included a table (as below) in the revised manuscript.

TDCs

Resolution

Conversion rate

Max. detection range

Disadvantages

Flash

coarse

high

small

vulnerable to PVT variations,

large chip area

Vernier

fine

low

moderate

large chip area, long dead-time

Cyclic Vernier

fine

very low

large

severe switching noise, high dynamic power consumption, large latency

GRO

fine

low

large

fast clock requied,

severe quantization error

TA

fine

high

large

high power consumption,

large gain required with extra calibration

Proposed I-TDC

fine

high

large

sensitive to PVT variations

Page 5, line 175. You write “Fig. 5 depicts the schematic diagram…”. Fig. 5 depicts the waveforms. So, please, put the figures in correspondence with the text. You had the same problem with Fig. 2 and Fig. 3.

-> We apologize for this confusion. Figures was placed in accordance with the text.

Page 9, Fig. 8. This figure describes the results of measurements, not the results of simulation, as I am supposing.

-> Thanks a lot for this comment. Certainly, the measured results were demonstrated in Fig. 8 (now Fig. 7 in the revised manuscript). We have corrected the caption.

Simplify the style and improve the grammar.

-> Thanks a lot for your advice.

Round 2

Reviewer 1 Report

In the revised manuscript, the author explained the principle of the decoding circuit, answered PVT related questions, and revised the rhetorical errors raised. I have no other questions to ask.

Reviewer 2 Report

  1. Although the author said that the output corresponds to digital codes and that the measurement was performed using a logic analyzer, the author discussed the analog value. If you get the digital codes from the proposed circuit with a 0.5-ns pulse, the threshold voltage of F1 must be smaller than 73 mV. On the other hand, if the threshold voltage is such a small value, detecting 56-ns, 111.5-ns, and 112-ns pulses is difficult because of the saturation. Describe the proposed circuit's resolution, LSB, and threshold voltage if you treat the output voltages as digital signals. Also, if you suppose putting an ADC for each F# output, this circuit is still a time-to-voltage converter.
  2. The author did not answer the advantages of dToF. Popularity is not an advantage. The question is why dToF is worth using for short-distance measurement in a home-monitoring LiDAR, though a quite precise TDC circuit is required.
  3. Other answers are clear.

Author Response

1. Although the author said that the output corresponds to digital codes and that the measurement was performed using a logic analyzer, the author discussed the analog value. If you get the digital codes from the proposed circuit with a 0.5-ns pulse, the threshold voltage of F1 must be smaller than 73 mV. On the other hand, if the threshold voltage is such a small value, detecting 56-ns, 111.5-ns, and 112-ns pulses is difficult because of the saturation. Describe the proposed circuit's resolution, LSB, and threshold voltage if you treat the output voltages as digital signals. Also, if you suppose putting an ADC for each F# output, this circuit is still a time-to-voltage converter.

-> (ans.) Thanks a lot for this valuable comment. We authors apologize for having not fully described the circuit operations. Below are given the detailed explanations and these are also included in the revised manuscript.

“For these measurements, two threshold voltages are selected, i.e., one (VH) is set to ~950 mV and the other (VL) is set to the LSB which is 37.5 mV. Then, the results (F1~F15) of the I-TDC are compared with VH, resulting in a coarse thermometer code that will be eventually converted to a binary code. Thereafter, a result lied in between VH and VL is selected among F1~F15, and is connected to a simple 5-bit ADC by using a switch, finally generating a fine-control code. As an example, Fig. 7(a) results in a coarse control code with 14 zeros. Then, F1 is selected and connected to the 5-bit ADC, producing a 5-bit fine code (i.e., 00010). In the case of Fig. 7(c), the coarse code has 6 ones for F1~F6, and 8 zeros for F7~F14. Then, F7 is selected to the ADC, generating a fine code (i.e., 00010). For the case of Fig. 7(f), the coarse code has 14 ones, indicating the maximum range distance. It is noted here that only a 5-bit ADC with 15 switches connecting F1~F15 can be utilized for this proposed I-TDC, hence enabling to avoid the undesired increase of power consumption.”

As described above, our primary goal in this work was to present a new methodology of using an integrator in a TDC circuit. And the feasibility of this idea was preliminarily checked by the measured output voltages which could then be used to analyze the range detection. Hence, we authors reckon that this work can be considered as a potential low-cost low-power TDC solution for indoor monitoring LiDAR sensors.

2. The author did not answer the advantages of dToF. Popularity is not an advantage. The question is why dToF is worth using for short-distance measurement in a home-monitoring LiDAR, though a quite precise TDC circuit is required.

-> (ans.) In general, dToF has pros (e.g., fast acquisition, no ambiguity, multiple echoes, digital dynamic range) and cons (e.g., reduced no. of pixels, large data volume). Particularly, its comparison with iToF reveals some characteristics as described below.

(1) dToF is well known as the best solution for range finder by adopting ‘stop watch’ approach, although a TDC circuit is required to measure the distance precisely. To the contrary, iToF is the best for 3D imaging by detecting phase shift.

(2) dToF uses SPADs (single photon avalanche diodes) or APDs (avalanche photodiodes), whereas iToF needs PMDs (photonic mixer devices).

(3) dToF emits VCSEL (or laser) output pulses with 0.2 ~ 5 ns pulse width, whereas iToF emits modulated sinewaves (20 ~ 100 MHz).

In short, we reckon that dToF is simple and easy to design, exploits very well-known optical devices such as APDs/SPADs, enables a single-shot measurement with fast acquisition, and needs no modulation scheme in transmitter. These would be the advantages of dToF for the purpose of short-range home-monitoring.

3. Other answers are clear.

-> (ans.) Thank you.

Reviewer 3 Report

  1. The description why your circuit is needed should be shortened. You did not do it.
  2. In comparative table you indicated that your circuit has has high sensitivity to PVT variations. This is a very negative characteristic of your circuit. Do you realize this?

Author Response

1. The description why your circuit is needed should be shortened. You did not do it.

->(ans.) According to the reviewer’s previous and current comments, we have shortened the 2nd paragraph in introduction referring to only one country, as below.

“In particular, home-monitoring LiDAR sensors have become urgently needed to help professional nurses take care of their dementia patients more efficiently. It is well known that many developed countries have been facing the surge of senior citizen population due to their advanced health systems [2], which naturally leads to the proliferation of senile dementia patients. In South Korea, the Ministry of Health-Welfare predicted that 10 % of senior citizens, i.e., one-million elders might suffer dementia in 2025. The Alzheimer Nederland Foundation anticipated that 13 % of senior citizens in Nederland, i.e., half a million elders might suffer dementia in 2040. However, these elders would frequently suffer from unpredictable falling accidents, thus demanding extremely high social costs for their medical care. In 2012, the statistics of falling accidents over 65 years old were disclosed in Canada, indicating that 50 % of the falling accidents occurred in houses and 60 % of these falling accidents were caused during walking, and that the hospitalization rates related to these accidents have increased rapidly.

2. In comparative table you indicated that your circuit has high sensitivity to PVT variations. This is a very negative characteristic of your circuit. Do you realize this?

-> (ans.) Thanks a lot for this comment. As commented, we admit that our circuit is sensitive to PVT variations because a flash TDC is exploited as a coarse-control circuit.

Round 3

Reviewer 2 Report

The paper’s scope entitled “A CMOS Integrator-based Clock-free Time-to-Digital Converter for Home-Monitoring LiDAR Sensors” is concerned with developing a time-to-digital converter (TDC) using integrators without a clock generator for indoor monitoring applications. The authors answered reviewers’ questions satisfactorily. I concluded that the paper should be accepted in the MDPI Sensors.  

Minor comments:

  1. Thanks to the authors’ answers, I understand the novelty of the proposed device as a proof-of-concept. However, the FOM and core area comparison in Table 2 seems unfair (with ADC vs. without ADC). It is just a recommendation, but it might be better to omit the FOM and core area rows in Table 2 when authors submit the published version. It may not affect the text. However, if the authors are sure these comparisons are required, it is also acceptable.